# The Effectiveness of a Foot and Mouth Disease Outbreak Control Programme in Thailand 2008–2015: Case Studies and Lessons Learned

**DOI:** 10.3390/vetsci5040101

**Published:** 2018-12-06

**Authors:** Terdsak Yano, Sith Premashthira, Tosapol Dejyong, Sahatchai Tangtrongsup, Mo D. Salman

**Affiliations:** 1Faculty of Veterinary Medicine, Chiang Mai University, Chiang Mai 50100, Thailand; vetjek@gmail.com or terdsak.yano@cmu.ac.th (T.Y.); sahatchai.t@cmu.ac.th (S.T.); 2Department of Livestock Development, Bangkok 10400, Thailand; sith.prem@gmail.com (S.P.); tosapol_palm@hotmail.com (T.D.); 3Animal Population Health Institute, College of Veterinary Medicine and Biomedical Sciences, Colorado State University, Fort Collins, CO 80523, USA

**Keywords:** cattle, Foot and Mouth Disease, elephant, lessons learned, outbreak control programme, pig, Thailand

## Abstract

Three Foot and Mouth Disease (FMD) outbreaks in northern Thailand that occurred during the implementation of the national FMD strategic plan in 2008–2015 are described to illustrate the lessons learned and to improve the prevention and control of future outbreaks. In 2008, during a FMD outbreak on a dairy farm, milk delivery was banned for 30 days. This was a part of movement management, a key strategy for FMD control in dairy farms in the area. In 2009, more than half the animals on a pig farm were affected by FMD. Animal quarantine and restricted animal movement played a key role in preventing the spread of FMD. In 2010, FMD infection was reported in a captive elephant. The suspected source of virus was a FMD-infected cow on the same premises. The infected elephant was moved to an elephant hospital that was located in a different province before the diagnosis was confirmed. FMD education was given to elephant veterinarians to promote FMD prevention and control strategies in this unique species. These three cases illustrate how differences in outbreak circumstances and species require the implementation of a variety of different FMD control and prevention measures. Control measures and responses should be customized in different outbreak situations.

## 1. Introduction

Foot and mouth disease (FMD) is caused by a virus in the family Picornaviridae and the genus Aphthovirus and is a highly contagious disease of cloven-hoofed animals. Seven serotypes of foot and mouth disease virus (FMDV) have been identified including serotype A, O, C, Asia 1 and South African Territories (SAT) 1, 2 and 3 [1]. Of these, Types A (1953) O and Asia-1 have been identified in Thailand. Species known to be susceptible to FMDV include cattle, sheep, goats, water buffaloes, elephants and other cloven-hoofed animals [2].

Since 1953, Thailand has been considered an endemic area for FMD [3]. Currently, the circulating serotypes of FMDV in Thailand are O-Myanmar 98 topotypes and A-SEA 97 topotypes; Asia 1 has not been detected since 1998. Most of the outbreaks have been reported in cattle with some outbreaks occurring in pigs [4,5].

FMDV transmission occurs by aerosols, direct contact with infected animals and contaminated fomites such as shoes, tires and equipment [6]. The virus can also survive in the human respiratory tract for longer than 24 h, making transmission of disease from humans to animals possible. Clinical signs of FMD in livestock include fever, vesicular lesions, erosions and ulcers in the mouth, on the muzzle, teats and coronary band. Clinical signs can result in reduced milk production in dairy cattle and weight loss in beef cattle and pigs. FMD causes high morbidity but usually low mortality, except in young animals [7]. 

Presence of FMD is an important barrier to trade; countries that are free of FMD do not import unprocessed beef and pork products from FMD-affected countries. The disease has become a serious impediment not only to efficient livestock production, however also to livestock export opportunities for Thailand.

According to 2015 data from the Department of Livestock Development (DLD), there were 4,916,632 cattle, 888,431 buffaloes and 9,886,897 pigs in Thailand [8,9,10]. Approximately 10% of those animals were located in the Northern part of Thailand that borders Myanmar and Laos. Animals in this area are susceptible to FMDV, which may enter Thailand with imported animals from neighboring countries. 

The Thai DLD is in charge of the prevention, control and possible eradication of FMD from Thailand. Control measures for FMD include animal movement restrictions, a vaccination programme, animal quarantine, environmental sanitary controls, outbreak investigation, serological surveillance and slaughtering of sick animals [11]. A national FMD strategic plan 2008–2015 was developed and applied throughout the country by the DLD. Vaccination, along with disease investigation and control, were the most important activities for FMD control in Thailand [12]. The goals were to reduce FMD incidence in Thailand and to establish an FMD-free zone using an effective vaccination programme. Preventive measures were used to strengthen disease protection in animals and on farms. Control measures were selectively employed depending upon local needs and conditions to minimise the impact of the disease. A national FMD control programme 2016–2023 was submitted to the OIE (World Organisation for Animal Health) for endorsement with the aim to be free from FMD by 2023 [13]. 

The aim of this paper is to describe the national FMD strategic plan 2008–2015, describe the FMD outbreak situation over this time period and discuss three selected outbreaks in northern Thailand that occurred during implementation of the strategic plan. These cases are presented to illustrate the effectiveness of the plan and describe lessons learned to improve the control of FMD in Thailand. 

## 2. Foot and Mouth Disease Control Measures in Thailand 2008–2015

The national strategic plan developed by the DLD included many measures that harmonised with the regional World Organisation for Animal Health (OIE) campaign in Southeast Asia and China (SEACFMD) [14]. The goals of the strategic plan were to decrease FMD incidence, to enhance biosecurity on livestock farms and to establish an FMD-free zone with vaccination, potentially in the eastern region. There was also an action plan for FMD outbreak response that followed the existing control and eradication plan for field veterinarians [15].

The preventive measures of the 2008–2015 national strategic plan were as follows: 

(1). Routine vaccination of cattle, buffaloes, goats and sheep. In dairy cattle, routine FMD vaccination was applied three times per year free of charge. The DLD trivalent vaccines, containing A (Thailand 118/87 and A Sakolnakorn/97), O (Thailand 189/87) and Asia 1 (Thailand/85), were distributed via dairy cooperative networks and vaccination of all dairy cattle was compulsory by their owners or farm veterinarians. Beef cattle and buffaloes were vaccinated twice annually, which was also free of charge. The percentage of trivalent vaccine coverage during fiscal year 2014 was 106.47% in the first round and 109.24% in the second round [16]. The coverage was over 100% due to the use of booster vaccine for young animals as per the instructions from the vaccine manufacturer.

(2). Program vaccination of pigs. Pigs were vaccinated according to their farm FMD vaccination programme, however vaccination was not compulsory. Farm owners were educated on the importance of vaccination for FMD. The DLD produced vaccine for pigs below the market price and the purchase of the vaccine was subsidised by the Thai government. The vaccine coverage in pigs was not recorded because FMD vaccine usage in pig farms was not required to report to DLD. 

(3). Individual animal identification and traceability system. The DLD implemented the National Livestock Identification and Registration System (NID) programme to identify cattle, buffaloes, sheep and goats in the country with NID ear-tags. Different colour ear tags designate the source of the animals, including imported animals, and DLD officers are able to trace animals from these tags. A traceability system using registration of farms, slaughterhouses and butchers was used in pork and beef production throughout the production chain.

(4). FMD outbreak contingency plan. DLD officers at the district, provincial and regional levels were trained and provided with a FMD contingency plan in the event of a FMD outbreak in their area. Emergency Disease Control units were set up to respond and to control outbreaks.

(5). Farm biosecurity improvement. Farmers in cattle and pig production units were educated on the implementation of good biosecurity practices on their farms as an important tool to prevent the introduction of FMDV. The farmers were trained regarding the Good Agriculture Practice (GAP) for livestock, including biosecurity, at least one time by DLD. Additional trainings about biosecurity could be performed by related organizations, such as a university.

An outbreak of FMD is suspected if at least one animal on the farm shows two or more clinical signs including fever, depression, salivation, lameness, vesicles in the mouth and/or on the coronary bands, teats or udder [17]. FMD control measures are undertaken immediately to stop the spread of the disease. The FMD control measures are (1) quarantine of the suspect premises, (2) tissue lesion sample submission to the laboratory, (3) outbreak area officially defined to strictly control animal movement, (4) animal check point set up, (5) ring vaccination, (6) outbreak investigation, (7) cleaning and spraying of disinfectant, (8) disposal of carcasses and infected materials and (9) public awareness [18]. Once FMD is suspected or an outbreak is reported, the local DLD authority orders confinement of the infected premises where the suspect case is located. The outbreak investigation team confirms appropriate intervention measures to eliminate FMD at the source. Until 30 days after the last FMD case, milk cannot be shipped and cattle or pigs cannot be sent for trade or slaughter without compensation. FMD control measures not only target infected farms, however also the movement of animals and products.

## 3. FMD Outbreaks in Northern Thailand 2008–2015

Livestock policy, including animal health, is enforced through nine DLD regional offices around the country. Northern Thailand is an important area for livestock production, especially pigs, dairy and beef cattle [8,9,10], and includes eight provinces—Chiang Mai, Chiang Rai, Lamphun, Lampang, Payoa, Prae, Nan and Maehongson under the Fifth Regional Livestock Office of the DLD (Figure 1). During 2008–2015, a total of 140 FMD outbreaks were reported in these eight provinces (Figure 2). The Lamphun province had the highest number of outbreaks, followed by Chiang Mai and Lampang. The dry season in Thailand is September to January and is the high-risk season for FMD (Figure 3). A total of 101 outbreaks occurred on cattle premises, 22 occurred on premises that had cattle and buffaloes, 6 outbreaks occurred on buffalo premises, 6 outbreaks on pig premises and 4 outbreaks on premises that had both cattle and pigs. Only one of the reported FMD outbreaks in Northern Thailand was linked to a premise with a combination of cattle, buffaloes and pigs, which occurred in Maehongson province in 2008.

The majority of outbreaks were caused by FMD virus serotype O (67/140, 47.86%) and 32/140 (22.86%) were caused by serotype A infection. The serotype of FMDV could not be identified in 41/140 outbreaks (29.29%) due to either a lack of samples or inappropriate sample submission. Disease investigation data revealed that 36/140 outbreaks (25.71%) were related to animal movements. A variety of causes were attributable to 38/140 (27.14%) of outbreaks and included human and equipment movement onto farms, contaminated animal feed and spread from slaughterhouses and live animal markets. For 66/140 outbreaks (47.14%), a source could not be identified. During the seven years of the national plan, the vaccination status of individual animals was not recorded.

## 4. FMD Outbreak and Control Measure Implementation

The 140 FMD outbreaks that were officially reported through the national animal disease reporting system in Northern Thailand varied in their characteristics and control measures. Most of the outbreaks occurred on cattle and buffalo farms, particularly backyard and smallholder farms. In contrast, only intensive pig farms reported outbreaks. According to the Thai Animal Epidemics Act B.E. 2499 (1956) and its revision B.E. 2542(1999), once an outbreak begins, both cattle and pig farmers need to report the events to the DLD. The district DLD officer is the first to be directly involved in the outbreak area, followed by a provincial team of technical people who are assigned the task of applying control measures. Finally, the regional DLD officer supports various resources to minimise the impact. 

The following case reports are from three interesting and unique FMD outbreaks in Northern Thailand during 2008–2015 that demonstrate distinct outbreak patterns and implementation of control measures that were applied to halt the spread of disease.

### 4.1. Case 1: FMD in Dairy Cattle Farms

Dairy production is the major livestock activity in Northern Thailand, particularly in Chiang Mai, Lamphun and Chiang Rai provinces. Dairy farmers are assembled in dairy cooperatives with a milk collection center. Farmers deliver their milk to the dairy center; the milk is then sent to the milk company factory via cooling trucks.

On 28 August, 2008, suspected FMD in dairy cattle was reported to the San Sai district DLD officer in Chiang Mai Province. At that time, nine animals on the affected farm, a small holder with an open-air tie stall barn, had developed clinical signs that were typical of foot and mouth disease. A total of 51 cattle were on the farm including calves, heifers and milking cows, and 24 (47.0%) developed typical signs of FMD over the 6-day period of 26–31 August. No deaths were reported during this outbreak. The animals on the affected farm had been vaccinated with trivalent (O, A and Asia1) FMD vaccine approximately 85 days pre-infection. The incidence of FMD cases by day is shown in Figure 4. 

According to the DLD report, FMD cases were reported in two other districts in Chiang Mai Province, (Mae-On and Sankamphang), five and six days before the detection of the first case in San Sai district. FMD cases in those districts were determined to be related to animal movement from Kanchanaburi Province. A total of 12 dairy cattle were moved from Thamaka district, Kanchanaburi province, to a dairy farm in Mae-on district, Chiang Mai province on 14 August, 2008. No animal was reported with FMD signs in this district prior to 14 August. Later, on 23 August, according the DLD records, all 12 cattle showed clinical signs that were suspicious of FMD. On 22 August, the DLD officer in the neighboring Sankamphang district also reported FMD in four dairy cows. FMDV, in our case example, was possibly introduced to the dairy farm in San Sai district by the animal truck that came onto the farm to deliver cattle on 16 August. The truck was likely not cleaned or disinfected after carrying the FMD infected cattle and also did not have the official documents for approved animal movement.

The DLD officer in this case applied eight control measures: (1) All infected animals were treated with antibiotics to control secondary infection. Gentian violet was used topically on lesions. (2) A Glutaraldehyde disinfectant was provided by the DLD to disinfect the entire farm area including floors, around the cattle houses, cattle resting area and roads on the farm. Disinfectant was used once a day to reduce the viral load on the farm. (3) The DLD officer worked together with the dairy cooperative to monitor the clinical signs of FMD that were reported by the other cooperative members. Milk volume on other farms was also monitored as any reduction recorded at the milk collection center could indicate FMD. (4) Cattle within a 5 kilometers radius of the outbreak farm were vaccinated with FMD trivalent vaccine. FMD titer was determined 7–14 days after vaccination to monitor the immune response. (5) The DLD officer authorised a quarantine of the infected animal(s) on the farm for 30 days after the last case occurred. Restocking was discouraged until the virus was no longer detectable on the premises. (6) Within 5 km around the affected district, all animal movement required approval by the DLD district officer. (7) For disease confirmation, samples from lesions and serum samples were submitted to the Regional Reference Laboratory for Foot and Mouth Disease in Southeast Asia (RRL-FMD) to identify the serotype of the agent. FMDV monitoring was performed on days 47, 84 and 115 post-infection (PI) to check the carrier status of the infected animals. (8) The committee of the dairy cooperative ordered the affected farm to stop delivering milk until 30 days after the last infected animal was detected.

Samples from vesicular lesions were collected to confirm the infection through virus isolation at the RRL-FMD in Nakhon Ratchasima, Thailand. FMDV serotype A was detected. Serum and whole blood samples from eight cows with vesicular lesions were collected on days 47, 84 and 115 PI. On day 115 PI, saliva and esophageal (EP) fluid samples were collected from 21 cows on the infected farm. Sixty sera were also collected from cattle on surrounding farms. The results showed that 62.5% (5/8) of the infected animals had titers of 12 Log2 or greater against FMDV serotype A, corresponding to the recent infection. One hundred percent (8/8) and 50% (4/8) of the infected animals also showed antibody titers that were greater than 10 Log2 against FMDV serotype O and Asia1, respectively, which were higher than those observed in the surrounding farms. No FMDV genetic material was detected in saliva and EP fluid that was collected during day 115 PI. This information indicated that the outbreak vaccination programme, movement restrictions and early disease reporting after the detection of clinical signs were effective strategies in the control of FMD on this dairy farm. 

Cattle on this farm had been vaccinated with FMD vaccine 85 days before the outbreak, indicating that the vaccine was not protective during this outbreak. Officers who investigate FMD outbreaks must emphasize vaccine quality, vaccine administration and maintaining cold chain. The monitoring of immune response after vaccination should be performed to assess vaccine quality. The vaccinator, particularly livestock volunteers in the community, should be trained and refreshed each year in vaccination technique. Lastly, cold chain procedures must be followed and should be documented during transit to protect vaccine efficacy from the vaccine factory all the way to the animal itself. 

### 4.2. FMD Case 2: FMD on a Pig Farm

On 25 October 2009, clinical signs that were suspicious of FMD on a pig farm in Ban Thi district Lamphun Province were reported to the DLD district officer. The district team investigated and collected samples that same day. The affected farm was a farrow-to-finish operation with two production sites. The first site had five open houses for breeder and nursery pigs. The second site, which housed the grower-to-finisher unit, was located approximately three kilometers away. None of the pigs at either site had been vaccinated against FMDV. Only the breeder and nursery units were affected with FMD. Two boars, 106 sows and 568 nursery pigs were raised at this site. Clinical signs were noted in 355 of 676 pigs (52.51%). Among the affected animals were 84 sows (79.25%) and 271 nursery pigs (47.71%). There were 62 deaths (9.17%) of which three were sows (2.83%) and 59 were nursery pigs (10.39%). All of the infected animals showed clinical signs including fever, anorexia, stomatitis, lameness and blisters on teats, coronary band, snout and mouth.

The outbreak began on 13 October 2009 and the last case was reported on 22 November. The DLD officer was on the farm on 25 October soon after the owner informed the DLD. Farm biosecurity practices were lacking and likely increased the risk of an outbreak. For example, manure and feed trucks moved freely on and off the premises without disinfection, visitors were allowed on the farm without special precautions, the disinfectant pool at the farm entrance was not used and the FMD vaccination programme had ceased more than two years prior.

A total of seven control measures were utilised in this case: (1) Supportive treatment including antibiotics, vitamins and minerals were given to clinically-ill animals to improve survivability. (2) A Glutaraldehyde disinfectant was sprayed once daily after routine cleaning. Disinfectant spray was also used during movement of pigs to the finisher farm. (3) Beef and dairy cattle in Ban Thi district were monitored during the outbreak. Cattle on four beef farms were sampled and sera were collected to test for FMD titers. (4) Four FMD vaccination cycles were carried out on the outbreak farm. When the owner suspected FMD on his farm, all sows and finishing pig in the finisher unit were vaccinated with DLD-FMD trivalent vaccine on 19 October 2009. The outbreak, however, became worse and the owner informed the DLD officer. On 27–28 October, the DLD officer recommended vaccination of the finishing pigs and nursery pigs that were older than two months. Booster vaccination of the pigs was performed two more times on November 4 and 18. Beef and dairy cattle within a five-kilometer radius were also vaccinated. (5) All pigs were quarantined on the farm until 30 days after the last case. However, finishing pigs that were vaccinated with FMD vaccine on 4 and 18 November and were kept in the breeder unit were required to move to the finisher unit before the end of the 30 days. They were removed on day 29 after the last case. The DLD team sprayed disinfectant on the truck that was used to transport the pigs from the breeder unit to the finisher unit. (6) All animal movement in Ban Thi district had to be approved by the DLD district officer, especially movement outside the area. The officer at the origin was required to check for clinical signs of FMD before approval and the officer at the destination re-checked for clinical signs upon arrival. (7) Tissue from affected animals was collected to confirm FMD. Serum was also collected to monitor the immune response of the pigs after vaccination. Probang samples were collected to gain further information about the carrier status of the suspected pigs.

On 26 October 2009, tissue samples from the coronary band and snout were collected and sent to RRL-FMD in Nakhon Ratchasima, Thailand. The results showed that FMDV serotype O was the cause of the outbreak. Antibody titers of 11 cattle samples on four beef farms in the area indicated that infection with FMDV had not occurred in those animals. Blood collection was performed every seven days, from 1 to 8 weeks after the first vaccination to monitor the immune response. The results showed that pigs that received a booster dose had higher titers within a week after vaccination. Pigs that were infected with FMDV had high titers for eight weeks.

Vaccination programmes and biosecurity are essential components in disease prevention on pig farms. This farm disregarded FMD vaccination for two years and increased the vulnerability of pigs through lowering the herd immunity, especially the breeders, on this farm. At the earlier stage of the outbreak, all pigs were vaccinated with DLD-FMD trivalent vaccine, however the situation became worse since it was too late to build sufficient immunity to protect the non-affected animals. This information indicated that vaccination during outbreak was not effective. When the outbreak occurred, the owner should have focused on the isolation of affected animals and a rigid disinfection procedure to reduce contact as part of building better herd immunity with less emphasis on vaccination. Furthermore, vaccination, if implemented, should be done from the pig house farthest away from the infected house to the pig house nearest to the infected house. It was also noticed that the farmer had ignored biosecurity measures to prevent the introduction and spread of FMDV to the farm by allowing contaminated vehicles and people to come onto the farm. These two reasons are the likely cause of the FMD outbreak on this farm. 

### 4.3. Case 3: FMD in an Elephant Refuge Holding Facility

On 28 October 2010, the RRL-FMD received an elephant tissue sample from the National Elephant Institute, Lampang Province, Thailand for FMD confirmation, and the Provincial DLD office was notified. The DLD team visited the resident camp of this elephant. The initial investigation revealed that the 45-year-old captive-bred female elephant had been lame since 15 October and was given Penicillin-Streptomycin (Pen-Strep LA^®^) and Phenylbutazone (Butasyl^®^) by the owner as the initial treatment. On 21 October, blisters around the trunk and nail margins were observed. The elephant was moved to the hospital at the National Elephant Institute, Lampang, Thailand for intensive treatment on 22 October. Foot and mouth disease was suspected at that time.

The elephant received standard wound care, and an FMD antibody titer was evaluated using an ELISA test. A tissue sample from the interdigital area was collected and sent to the RRL-FMD in Nakhon Ratchasima, Thailand for subtyping using a PCR technique. Within a month, all wounds had healed and the elephant was returned to its resident camp.

The disease investigation indicated that the elephant had previously worked in a camp where tourists can watch elephant shows and ride elephants and cattle. Before the disease developed, the camp was flooded for one and a half days and the cow barn floor was covered by river water that was approximately 2-feet deep. One cow stood in the flooded barn and later developed lameness. The cow was taken off work status and was kept separately in the resting area starting 8 October.

On 15 October, the elephant in this case became lame, and three days later she was suspended from work and was rested in the barn that was close to the sick cow resting area. The cow was higher than the elephant and on a slope, therefore either secretions and/or waste products from the cow could flow though the elephant resting area. The elephant was fed on the ground, thus increasing the chance of contamination of food. Three days later, blisters and pus were found at the interdigital and nail margin areas of the elephant. Erosion wounds were also observed on the trunk and mouth, causing a decrease in appetite. The incubation period of FMD is 2–14 days, thus this elephant was likely infected with FMDV from the infected cattle [19].

Disease investigation was initiated after typing enzyme-linked immunosorbent assay (Typing ELISA) results confirmed that the elephant had FMD type O. However, molecular characteristic of the virus and R-values were not performed because of the small amount of sample that was collected. Additional tissue samples could not be collected because of the late recognition of the infection. The elephant was most likely exposed to the virus via food contaminated with secretions from the sick cow. 

The investigation team was not sure whether other animals were infected, so investigation that was based on a serology was planned. Blood and probang samples were collected from the four lame cows and 11 other cows in the same area. The results of four cow-blood samples showed that the lame cows had high antibody titers against FMDV serotype O and A. Only one cow showed an antibody against serotype A that was higher than serotype O and Asia1. The results of 11 cow-blood samples that were raised in the same area showed that nine samples had high antibody titers against FMDV serotype O, A and Asia1, while two samples showed low antibody titers against FMDV serotype O and A, however high antibody titers against FMDV serotype Asia1. All of the cows were also positive on the non-structure protein test (NS-test). None of the probang samples showed a positive result for FMDV. Blood samples were collected for serological testing from another 17 elephants in the same elephant camp and a nearby camp. The results were all negative for antibody titer to FMDV serotype O, A and Asia 1 and the NS-tests were also all negative.

Blood samples from the infected elephant were collected once a month for three months to evaluate the antibody titers for FMDV serotypes O, A and Asia 1. The results showed no titer in the first month and a slight increase in the second and third months. Antibody against serotype O was higher than it was for serotypes A and Asia1. The NS-test was positive for five months after clinical signs first began.

In this case, the infection was recognized at a late stage because the clinical signs of FMD, although present, were not recognized. The disease was detected while the elephant was being treated in another location. The spread of infection, however, did not occur. When the DLD team visited the elephant camp, FMD vaccination had been given to the beef cattle in the camp. Elephants there were not vaccinated because FMD in elephants is very rare. Cattle and elephants around the affected camp were monitored for clinical signs. There were no more infected animals in that area. In this case, the FMD infection in the elephant was unusual; elephant practitioners were educated about the clinical signs of FMD disease and how to prevent its spread. 

Even though FMD in elephants is very rare, the keeping of elephants with FMD susceptible hosts, such as cattle, can cause FMDV infection in elephants. Tourist places with a FMD susceptible host should apply FMD prevention measures, such as vaccination programmes and biosecurity, just as they do on livestock farms. Clinical signs monitoring and early reporting are additional control measures that can prevent the spread of FMDV. Moreover, non-farming personnel who are not familiar with FMDV should be trained on the recognition, prevention and control of infectious diseases such as FMDV. 

## 5. Discussion and Conclusions

A global FMD control strategy was established in 2012 by the joint work of FAO and OIE and aimed to limit FMD in endemic areas, particularly in developing countries. The FMD Progressive Control Pathway (PCP-FMD) and the OIE Performance of the Veterinary Services (PVS) Pathway are used to improve disease prevention and control not only for FMD, however also for other major diseases [20]. The SEACFMD campaign has focused on FMD eradication from Southeast and East Asia. The FMD control strategies framework, categorised as technical, advocacy and coordination, is being implemented in member countries [21]. The economic benefits of FMD freedom in Thailand demonstrated a predicted benefit-cost ratio of 3.73:1 without pork export and 15:1 with additional export [22].

In northern Thailand, the FMD outbreaks gradually increased from 2012 to 2015. This increase may be related to the improved DLD disease surveillance system as well as encouragement of farmers to report disease outbreaks to DLD officers. Lamphun province had the most FMD outbreaks in the region. A total of 48 outbreaks occurred in 2015, and every province under the control of the Fifth Regional Livestock Office had an outbreak. Most of the outbreaks occurred from September through January, which is the dry season in Thailand. September to October is the changing period between the wet and dry seasons. It could be the risk period, inducing stress in the animals and making them more susceptible to infection.

Each disease outbreak has its own pattern and unique characteristics; the three case examples presented here demonstrate the value of flexibility in the implementation of procedures to control FMD. 

The key disease preventative measures in Thailand are immunization, a traceability system, contingency planning and farm biosecurity. In an FMD outbreak, strict animal quarantine and movement restrictions, ring vaccination, cleaning and disinfection, carcass disposal, disease investigation with sample submission and public awareness are major control measures. The PCP-FMD and PVS have contributed to improving the effectiveness of these control measures [15].

All disease control measures can be implemented to control the spread of disease, however some may be more suitable than others depending on the context of the outbreak. In dairy cattle, milk delivery restriction from infected farms can be the first control measure implemented after the infection is recognised, limiting the spread of virus within the dairy community. As of 2015, the new Animal Epidemics Act empowers veterinarians to prohibit the movement of milk to fulfill movement control. On pig farms, animal movement restriction and disinfection can be conducted in parallel to reduce the spread of virus. In addition, personal biosecurity is a critical measure to limit virus transmission. In our pig outbreak example, the potential transmission route of FMDV introduction to the farm was a manure truck freely coming into the farm without disinfection. People, including farm staff or officers who work on FMD outbreak farms, must remove organic material through hand washing or showering and should clean outerwear before handling susceptible animals [23]. In the elephant case, disease investigation to define the source could be the high impact control measure to prevent the spread of FMDV.

Cleaning and disinfection are important control measures to reduce the microorganism load on the premise and prevent disease spread. Disinfectants should be selected to decontaminate the FMD affected farm, including persons, vehicles and fomites that could be leaving the farm. In Thailand, the DLD provides Glutaraldehyde to local DLD officers for disease control and they also use it in FMD outbreak control cases. Glutaraldehyde is effective biocide against bacteria and virus, however it is considered a carcinogen and toxin to aquatic organisms. The repeated long-term exposure increased the toxicity of Glutaraldehyde [24]. Personal protective equipment should be applied when using it and appropriate pretreatment of wastewater before release into the environment should be performed. Other appropriate disinfectants that effectively eliminate FMDV include Citric acid, Sodium carbonate, Sodium hydroxide, Iodophore or Potassium Peroxymonosulfate [25,26]. 

Molecular characteristics of the viruses and R-values, which were provided by the reference laboratory, indicated more information about the outbreak and helped to improve understanding of the spread of the outbreak, even in vaccinated animals. The molecular characteristic of the viruses also provided information about the strain of FMDV for vaccine manufacturing. In Thailand, trivalent FMD vaccine (serotype O, A and Asia 1) is produced, even though FMDV serotype Asia 1 has not occurred in Thailand since 1998. The DLD still provides trivalent FMD vaccine for pigs, cattle and buffalo farmers because outbreaks of FMDV serotype Asia 1 have occurred in the neighboring country of Myanmar [27]. To protect animals from FMDV by using vaccine, animals need to have immunity against all FMDV serotype that can possibly infect them. The samples with positive antibody titer to Asia 1 were likely the result of previous vaccination. 

Measures that are presented in the national FMD strategic plan can be implemented by improved engagement of animal and farm owners in the control programme. Recognition of the hidden threats of disease, delivered to farmers via risk communication, should encourage them to notify DLD officials of a disease outbreak and participate in control during the early stages [28]. Owners impacted by an FMD outbreak may increase the success rate of the FMD outbreak control. Because the economic impacts of disease outbreak are serious issues for farm owners and occur not only during outbreaks, however also with the control measures, risk communication should stress the loss of culture and livelihoods of farm owners that are impacted by an FMD outbreak.

The key activities of FMD prevention and control should continue and should be updated in order to push for better disease control achievement. From 2016 onwards, the timeline of performance indicators for key activities of FMD control was set up by the DLD. The activities are composed of a National Animal Identification System (NID), movement controls at the border, vaccination, risk-based surveillance, biosecurity enhancement and public awareness. These activities were proposed to OIE in 2015 [29].

The approach of the DLD officers during FMD outbreak control measures can have a major effect on the perception of farm owners. Friendly and open officers can increase the farm owners’ acceptance of their role. Respect for the owner’s opinions and ideas can break down the barrier between DLD officers and animal owners. In this way, during outbreaks, DLD veterinarians and officers could be a trusted information resource for risk communication and education about control [30].

One health approach, particularly collaboration, transdisciplinary and communication, can be applied in FMD control. Working together with the affected communities in outbreak controlling could create better understanding among officers and communities. In disease investigation, the FMDV spreading among livestock, wildlife and human should be identified to inform the necessary prevention measures in the affected communities. 

In the three sample cases, DLD officers worked closely with cattle and farmers and staff in the elephant camp. They shared ideas and chose practical solutions that the animal owners were able to perform. Effective collaboration between owners and officers and good biosecurity management are the keys to achieving greater success in outbreak control and prevention in the future.

The participation of the local community can reinforce outbreak control, especially on dairy farms. In Thailand, Dairy Cooperatives are where farmers collaborate on milk production, marketing and heath monitoring activities. An FMD outbreak is an important event that the cooperative must respond to immediately. Several FMD control measures can be implemented through the cooperative, particularly the measures which need the acceptance of farm owners.

The establishment of the future national FMD control strategic plan should include stakeholders that impact the success of FMD outbreak control. Pig, beef and dairy cattle farmers, animal traders and animal production companies are the main stakeholders that should be engaged to enhance the effectiveness of an FMD control strategic plan.

## Figures and Tables

**Figure 1 vetsci-05-00101-f001:**
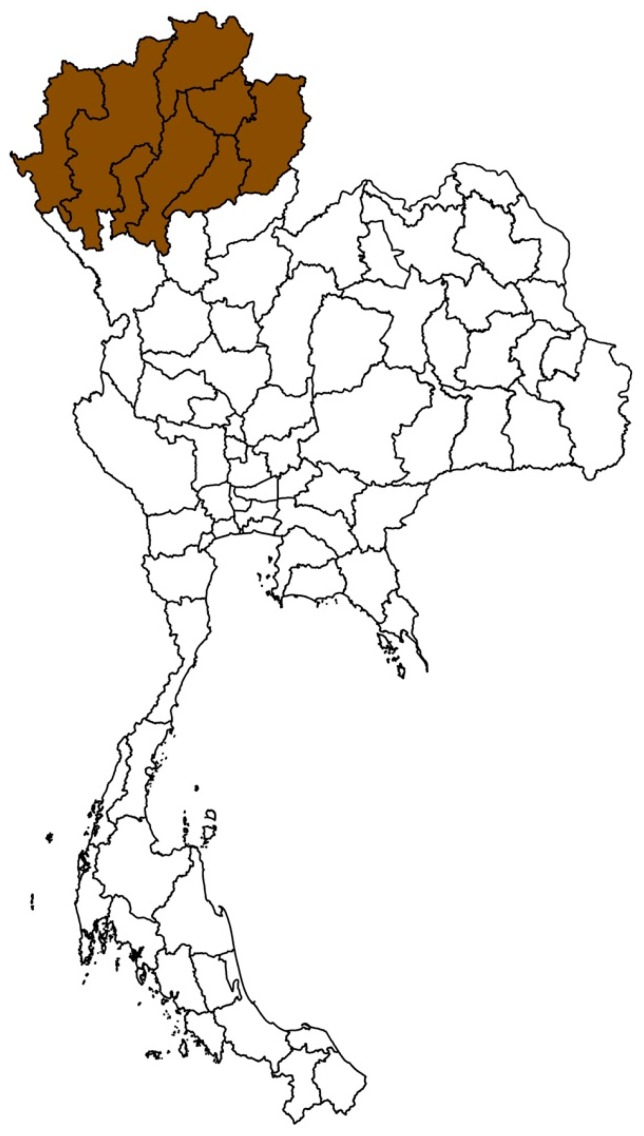
Map of Thailand highlighting eight provinces in northern Thailand under the Fifth Regional Livestock Office of the DLD (Department of Livestock Development) including Chiang Mai, Chiang Rai, Lamphun, Lampang, Payoa, Prae, Nan and Maehongson.

**Figure 2 vetsci-05-00101-f002:**
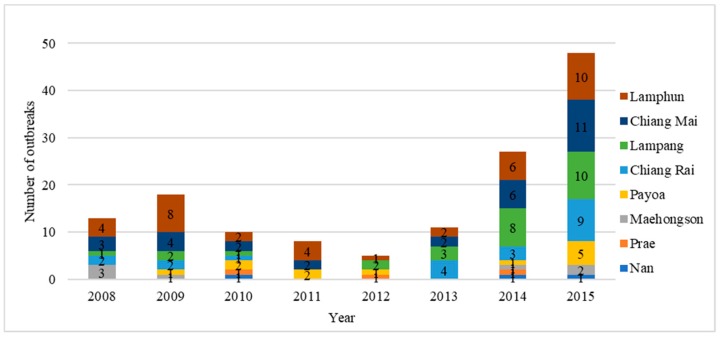
The Number of FMD (Foot and Mouth Disease) outbreaks in eight provinces in Northern Thailand, 2008–2015. The number in each colour box represents the number of outbreaks in each province.

**Figure 3 vetsci-05-00101-f003:**
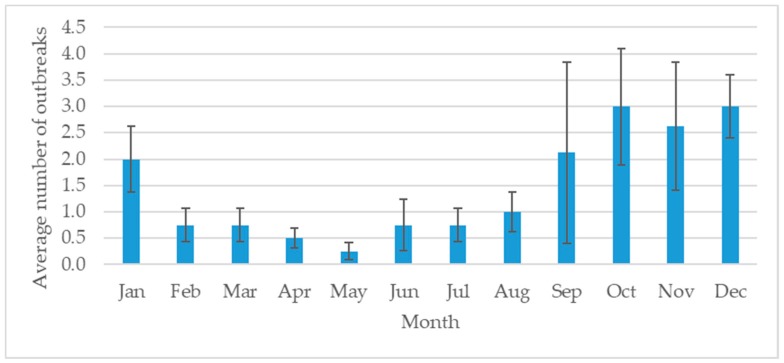
The average number of FMD outbreaks in eight Northern Provinces of Thailand, by month during 2008–2015.

**Figure 4 vetsci-05-00101-f004:**
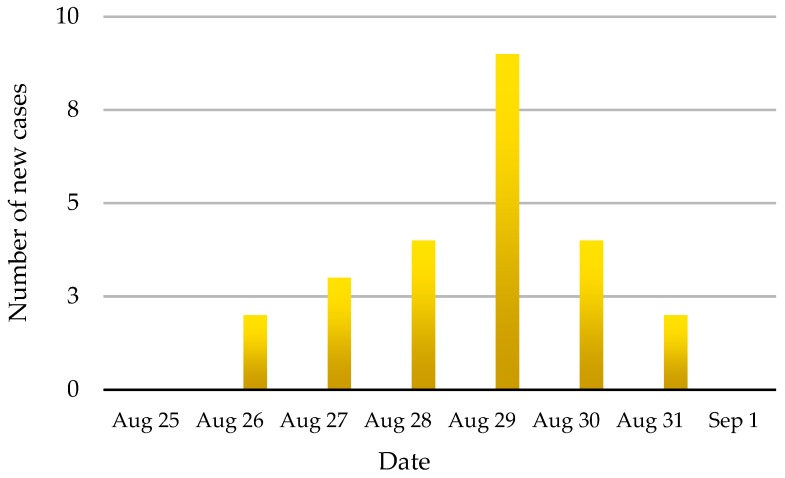
The incidence of FMD cases on an infected dairy farm in Chiang Mai Province during 2008. A total of 24 of 51 cattle showed pathognomonic signs of FMD.

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
