# Peer review of "The Effectiveness of a Foot and Mouth Disease Outbreak Control Programme in Thailand 2008–2015: Case Studies and Lessons Learned"

_vetsci, 2018, doi:10.3390/vetsci5040101_

Round 1

Reviewer 1 Report

I would suggest a few points for clarification.

Line 88. I assume you mean "usage" as "coverage" would imply that ALL animals were vaccinated at least once.

Line 105. I would suggested the use of "educated" for information supply, or "trained" where this involved some teaching, rather than "oriented". I t would also be good to specify how often, or when  such training took place.

Line 127. The absolute number of outbreaks is not informative unless we have details of the denominator (number of herds available to be infected). If this is available, it would be very helpful.

Figure 1. Please clarify if these are the only infected provinces.

Line 233. Would you suggest that cold chain procedure should be documented during transit, as well as followed.

Author Response

Dear Reviewer,

I would like to thank for your suggestion and recommendation. 

The following are my responses, point by point.

If you need more clarification, please let me know.

I appreciate to respond.

Line 88. I assume you mean "usage" as "coverage" would imply that ALL animals were vaccinated at least once.

-       Yes, your understanding is correct. DLD vaccinated cattle 3 times a year (around 4 months per round). They attempted vaccinate all cattle and some animals, especially young animals were vaccinated 2 times for booster within one round. So, the percentage of vaccine usage is over 100%

Line 105. I would suggested the use of "educated" for information supply, or "trained" where this involved some teaching, rather than "oriented". It would also be good to specify how often, or when  such training took place.

-       I have change and added more information as your suggested in line 106-110.

Line 127. The absolute number of outbreaks is not informative unless we have details of the denominator (number of herds available to be infected). If this is available, it would be very helpful.

-       The number of FMD outbreaks in this paper referred from DLD passive report. They count one outbreak as one area that they could detect. In one outbreak might include more than one herd or one farm. So, the number of infected farms could not be extracted. Moreover, total number of vulnerable herds in the outbreak area was not reported. For my opinion, the absolute number can be compared among the difference years and it could indicate other results such as the success of prevention measure in the next year.

Figure 1. Please clarify if these are the only infected provinces.

-       This map showed all provinces in the Fifth Regional Livestock Office of the DLD (Thailand has 9 Regional Livestock Office of the DLD). I also added the sentence to clarify reader’s understanding.

Line 233. Would you suggest that cold chain procedure should be documented during transit, as well as followed.

-       I added this information in line 244-245

Reviewer 2 Report

Yano et al. provide an interesting look at the FMD situation in Thailand. They present three case studies and discuss the country’s FMD eradication plan. This is a well-written paper with pertinent information and very valuable insights that should find a broad audience. Some pieces of the puzzle, however, seem to be missing, and should be added to the paper.

line 84: Routine vaccinations are carried out using a domestically produced trivalent vaccine containing FMDV O, A and Asia 1. What virus strains are contained in the vaccine? How closely related are they to the strains currently circulating in country?

line 88: I don’t understand the math here. Vaccine coverage simply is the count of animals vaccinated divided by the size of the target population. How can booster vaccinations push that over 100%?

line 92: What is the vaccine coverage in pigs?

line 117: Is there any compensation paid for the losses incurred from movement restrictions?

line 145: The error bars for all points in Figure 3 are the exact same size. I assume that is a mistake.

line 196: A truck that had previously carried FMD-infected cattle is mentioned as the possible source of this outbreak. Is that conjecture or is it know that the truck in question had actually carried infected cattle? Was any sequencing done for any of the outbreaks discussed in the paper? Is there any molecular evidence for the epidemiological links that are being suggested?

line 228: Several factors that can reduce vaccine efficacy are discussed, but a very important one is missing – does the vaccine match the circulating strains?

line 239: “affected farm”, not “effected”

Author Response

Dear Reviewer,

I would like to thank for your suggestion and recommendation. 

The following are my responses, point by point.

If you need more clarification, please let me know.

I appreciate to respond.

line 84: Routine vaccinations are carried out using a domestically produced trivalent vaccine containing FMDV O, A and Asia 1. What virus strains are contained in the vaccine? How closely related are they to the strains currently circulating in country? 

-       The virus strains in Vaccine consist of 

o   A          - Thailand 118/87 and A Sakolnakorn/97

o   O         - Thailand 189/87

o   Asia 1   -Thailand/85

         I added this information in line 86 

-       The vaccine strain is closely related with the circulating FMDV. OIE Regional reference laboratory for FMD in South East Asia (RRL-FMD-SEA) reported that in 2017 r-value of vaccine strain is good (0.4-1.0=100%) for samples from Thailand. (http://www.rr-asia.oie.int/fileadmin/Regional_Representation/Programme/O_others/2017_Dossiers_prep_Tokyo/1-2_W._Linchongsubongkoch_Develop_and_Current_Situation_FMD.pdf). 

However, the vaccine match should be monitored every year. I have also discussed this issue in the discussion line 424-433. 

line 88: I don’t understand the math here. Vaccine coverage simply is the count of animals vaccinated divided by the size of the target population. How can booster vaccinations push that over 100%?

-       DLD vaccinated cattle 3 times a year (around 4 months per round). They attempted vaccinate all cattle and some animals, especially young animals were vaccinated 2 times for booster within one round. So, the percentage of vaccine usage is over 100%

line 92: What is the vaccine coverage in pigs?

-       The pig farmers do not report FMD vaccine usage in their farm because FMD vaccination in pig farm is not mandatory. So, DLD does not record FMD vaccine usage in pig farms. I added this issue in line 95-96 to clarify your question.

line 117: Is there any compensation paid for the losses incurred from movement restrictions?

-       Compensation does not be applied with animal restriction. It is applied only with stamping out , according to the Animal Epidemics Act 2015. I added the word “without compensation” in line 127 to make more clear understanding.

line 145: The error bars for all points in Figure 3 are the exact same size. I assume that is a mistake.

-       I have revised and made the error bar as SEM of each bar. 

line 196: A truck that had previously carried FMD-infected cattle is mentioned as the possible source of this outbreak. Is that conjecture or is it know that the truck in question had actually carried infected cattle? Was any sequencing done for any of the outbreaks discussed in the paper? Is there any molecular evidence for the epidemiological links that are being suggested?

-       This is the possible source of outbreak that indicated in outbreak investigation report, which done by DLD officer. However, DLD did not show molecular evidence that link between truck and the outbreak because they could not collect any sample from the truck. 

-       After the outbreak and DLD officer indicated that the truck is possible source of the virus, the applied animal movement restriction to that area as I have mention in line 218-221 as followed

“5) The DLD officer authorised a quarantine of the infected animal(s) on the farm for 30 days after the last case occurred. Restocking was discouraged until the virus was no longer detectable on the premises. 6) Within 5 km. around the affected district, all animal movement required approved by the DLD district officer.”

line 228: Several factors that can reduce vaccine efficacy are discussed, but a very important one is missing – does the vaccine match the circulating strains?

-       Unfortunately, we cannot get vaccine matching information from RRL-FMD-SEA. So, we cannot say that vaccine match with the circulating strains or not. However, I have discussed this issue in line  424-433

line 239: “affected farm”, not “effected”

-       I have changed according your suggestion

Best regards, 

Terdsak Yano 

Reviewer 3 Report

This is a well conducted investigation of FMD outbreak in Northern Thailand. The paper is well written, and I rate it highly for publication.

The only recommendation that I would add is that manuscript should highlight the “one health” surveillance approach that was used to simultaneously investigate the outbreak in cattle, elephants and humans, which informed the necessary prevention measures in the affected communities.   The mention of the  "one health" approach will  also alert a wider readership and referencing of the important findings of the outbreak investigation.  

Author Response

Dear Reviewer,

I would like to thank for your suggestion and recommendation.

I agree with you that adding "One health approach" to the manuscript will engage wider readers.

So, I added the paragraph that related with one health in the discussion line 454-458 as you recommended.

Please se those paragraph.

" One health approached, particularly collaboration, transdisciplinary and communication, can be applied in FMD control. Working together with the affected communities in outbreak controlling could make better understanding among officers and communities. In disease investigation, the FMDV spreading among livestock, wildlife and human should be identified to inform the necessary prevention measures in the affected communities."

If you need more clarification, please let me know.

I appreciate to respond.

Best regards,

Terdsak Yano